# Reservoir Computing with Delayed Input for Fast and Easy Optimisation

**DOI:** 10.3390/e23121560

**Published:** 2021-11-23

**Authors:** Lina Jaurigue, Elizabeth Robertson, Janik Wolters, Kathy Lüdge

**Affiliations:** 1Institute of Theoretical Physics, Technische Universität Berlin, Hardenbergstr. 36, 10623 Berlin, Germany; 2Deutsches Zentrum für Luft- und Raumfahrt e.V. (DLR), Institut fur Optische Sensorsysteme, Rutherfordstr. 2, 12489 Berlin, Germany; Elizabeth.Robertson@dlr.de (E.R.); janik.wolters@tu-berlin.de (J.W.); 3Institut für Optik und Atomare Physik, Technische Universität Berlin, 10623 Berlin, Germany; 4Institute of Physics, Technische Universität Ilmenau, Weimarer Str. 25, 98693 Ilmenau, Germany; kathy.luedge@tu-ilmenau.de

**Keywords:** reservoir computing, time series prediction, performance optimisation

## Abstract

Reservoir computing is a machine learning method that solves tasks using the response of a dynamical system to a certain input. As the training scheme only involves optimising the weights of the responses of the dynamical system, this method is particularly suited for hardware implementation. Furthermore, the inherent memory of dynamical systems which are suitable for use as reservoirs mean that this method has the potential to perform well on time series prediction tasks, as well as other tasks with time dependence. However, reservoir computing still requires extensive task-dependent parameter optimisation in order to achieve good performance. We demonstrate that by including a time-delayed version of the input for various time series prediction tasks, good performance can be achieved with an unoptimised reservoir. Furthermore, we show that by including the appropriate time-delayed input, one unaltered reservoir can perform well on six different time series prediction tasks at a very low computational expense. Our approach is of particular relevance to hardware implemented reservoirs, as one does not necessarily have access to pertinent optimisation parameters in physical systems but the inclusion of an additional input is generally possible.

## 1. Introduction

Reservoir computing (RC) is a machine learning method that is particularly suited to solving dynamical tasks [1]. It was introduced as a way of using recurrent networks for machine learning but circumventing the costly training of the network weights [2]. The main principle underpinning reservoir computing is that the reservoir projects the inputs into a sufficiently high dimensional phase space such that it suffices to linearly sample the response of the reservoir in order to approximate the desired target for a given task. For this to work, the reservoir must fulfil certain criteria: the response to sufficiently different inputs must be linearly separable, the reservoir must be capable of performing nonlinear transforms, and the reservoir must have the fading memory property [2]. However, even when these criteria are fulfilled, the performance depends greatly on the dynamics of the reservoir. Hence, in the past two decades a lot of research in the reservoir computing community has focused on the optimisation of the reservoir parameters [3,4,5,6,7,8,9]. Furthermore, the optimisation of the reservoir is a task-specific problem [1,10,11,12] and a universal reservoir, which performs well in a range of tasks, remains elusive.

In a recent paper [13], the authors aim to eliminate the issue of hyperparameter optimisation altogether by removing the reservoir. Their approach essentially takes the well-known nonlinear vector autoregression (NVAR) method, uses a less parsimonious approach to filling the feature vector, and adds Tikhonov regularisation. However, the method of [13] trades the optimisation of the reservoir hyperparameters for the optimisation of the feature vector elements and it cannot be asserted that the latter is generally less costly. Furthermore, one of the main factors driving research into reservoir computing forward is the possibility for hardware implementation [14,15,16,17,18,19], which is impractical when the reservoir is absent.

In this contribution we demonstrate a new approach that reduces the need for hyperparameter optimisation and is well suited to boosting the performance of physically implemented reservoir computers. Specifically, we show that, by adding a time-delayed version of the input for a given task, the performance of an unoptimised reservoir can be greatly improved. We demonstrate this by using one unaltered reservoir to perform six different time series prediction tasks. In each case the only optimisation parameters are the delay and input strength of the additional delayed input. The aim of this work is not to achieve the best possible performance, but rather to demonstrate that reasonable performance can be achieved for various tasks using the same reservoir and at a very low computational cost.

Using time-delayed input is a common approach for adding memory to feedforward networks [20,21,22,23] and is the basis of statistical forecasting methods [21,24]. However, despite the simplicity of this idea, to the best of our knowledge, time-delayed inputs have not been widely used to optimise the performance of reservoir computers. This may be because the focus has been on constructing reservoirs that have the necessary memory to perform a given task [1]. One study in which time-delayed inputs have been used to improve the performance of a time series prediction task is [25]. However, in [25], the manner in which the time-delayed input was constructed assumed that the memory requirements of the task monotonically decrease with increasing steps into the past and did not allow for the input scaling of the delayed input to be varied as a free parameter.

Our results are of particular relevance to the hardware implementation of reservoir computing, because in physical systems one does not always have access to the relevant hyperparameters necessary for optimisation of the task-dependent performance but it should always be possible to add an additional input.

## 2. Methods

In the following, we describe the reservoir computing concept, the model for the reservoir that we use, our proposed time-delayed input method, and the benchmarking tasks that are used to test our approach.

### 2.1. Reservoir Computing

In reservoir computing, the reservoir, which at this point can be treated as a black box, is fed an input and the response of the system is sampled a number of times. The responses are then linearly combined to approximate the desired output (see Figure 1a). The linear output weights are trained via linear regression, typically using Tikhonov regularisation or regularisation by noise [1]. A variant of reservoir computing, that is of particular relevance for hardware implementation, is time-multiplexed reservoir computing using only one nonlinear element [26]. In this scheme both the injection of the data into the reservoir and the filling of state matrix $\underline{\underline{S}}$ occur sequentially. Typically, a mask is applied to the input data in order to diversify the response of the reservoir to the input. In the training phase, the reservoir is fed a sequence of training data of length $K_{\mathrm{tr}}$. A mask of length $N_{v}$ is applied to each element of the training data, where $N_{v}$ corresponds to the readout dimension (i.e., the number virtual nodes). Hence, there are $N_{v}K_{\mathrm{tr}}$ time-multiplexed inputs that are sequentially fed into the reservoir. The corresponding state matrix, which has the dimensions $K_{\mathrm{tr}}\times \left(N_{v}+1\right)$, is filled row by row with an additional bias term of 1 at the end of each row. The training step is then to find the $\left(N_{v}+1\right)$ dimensional weight vector $\underline{W}$ that best approximates
(1)$$\underline{\hat{o}}\approx \underline{\underline{S}}\cdot \underline{W},$$
where $\underline{\hat{o}}$ is the vector of $K_{\mathrm{tr}}$ target outputs. The solution to this linear problem is given by
(2)$$\underline{W}={\left({\underline{\underline{S}}}^{T}\underline{\underline{S}}+\lambda \underline{\underline{I}}\right)}^{-1}{\underline{\underline{S}}}^{T}\underline{\hat{o}},$$
where λ is the Tikhonov regularisation parameter and $\underline{\underline{I}}$ is the identity matrix.

#### Error Measure

To quantify the performance of the reservoir computer we use the normalised root mean squared error (NRMSE), defined as
(3)$$\mathrm{NRMSE}=\sqrt{\frac{\sum_{k^{\prime }=1}^{K_{o}}{\left({\hat{o}}_{k^{\prime }}-o_{k^{\prime }}\right)}^{2}}{K_{o}\mathrm{var}\left(\underline{\hat{o}}\right)}},$$
where ${\hat{o}}_{k^{\prime }}$ are the target values, $o_{k^{\prime }}$ are the outputs produced by the reservoir computer, $K_{o}$ is the length of the vector $\underline{\hat{o}}$, and $\mathrm{var}\left(\underline{\hat{o}}\right)$ is the variance of the target sequence.

### 2.2. Reservoir Model

To investigate the effect of delayed input on a physically implemented reservoir computer, we model a physical system that is inspired by optical delay line reservoirs [27,28]. Delay line implementations have shown promise due to high throughput speeds [29]. However, complex network connectivity, achieved via the introduction of multiple delays, represents a significant experimental hurdle, or requires opto-electrical conversion of the signal for electronic storage, thereby forgoing the advantages of an all-optical implementation. Recent developments in optical quantum memories with high bandwidth [30] and high capacity [31] allow for the on-demand storage and retrieval of optical pulses and thus the implementation of delays of arbitrary length, limited only by the coherence time of the optical memory, which can reach up to one second [32]. The reservoir model described below models a physical optical system including the optical memory for the reconfigurable and arbitrary coupling of the injected information (modeled as memory cells with input and output coupling), a nonlinear element (modeled as a semiconductor optical amplifier), and a short delay line, whose purpose is not for introducing delay, but to recouple existing information back into the optical system. A sketch of the envisaged setup is shown in Figure 1b. A time-multiplexed input is fed through a nonlinear element and then stored in the memory cells (described with the index *n*) with a certain input topology $K_{\mathrm{in}}^{n}$. Combinations of the memory cells are partially read out with the output topology $K_{\mathrm{out}}^{n}$ and finally coupled back into the nonlinear element. Since the write and read-out process repeats in time, it is possible to realise time-varying read and write topologies. We describe this by adding the index m=k(modM), where *M* is the period within which the coupling sequence repeats, giving the coupling matrix elements $K_{\mathrm{in}}^{mn}$ and $K_{\mathrm{out}}^{mn}$. The map describing this process is given by the following. Let $x_{n}\left(k\right)$ be the state of the $n^{th}$ memory cell at time step *k*. The next time step is then given by
(4)$$x_{n}\left(k+1\right)=K_{\mathrm{in}}^{mn}G\left(Kx_{out}\left(k\right)+J\left(k+1\right)\right)+K_{\mathrm{nin}}^{mn}\left(1-K_{\mathrm{out}}^{mn}\right)x_{n}\left(k\right),$$
where
(5)$$x_{out}\left(k\right)=\sum_{n=1}^{N}K_{\mathrm{out}}^{mn}x_{n}\left(k\right).$$
$G\left(x\right)$ is the function describing the nonlinear element, and the matrices $K_{\mathrm{in}}$, $K_{\mathrm{nin}}$, and $K_{\mathrm{out}}$ describe the (possibly time-varying) coupling into and out of the memory cells. The value of *K* describes the percentage of the output $x_{out}\left(k\right)$ that is coupled back into the nonlinear element and $J\left(k\right)$ is the input. The coupling matrices have the dimensions *M*x*N*, where *N* is the number of memory cells. Note that the index m=k(modM) depends on the time step k. For each iteration one row of the coupling matrices determines which memory cells are written into and which are read out of. $K_{\mathrm{in}}$ gives the write sequence and $K_{\mathrm{out}}$ the out-coupling sequence. These two matrices contain values from zero to one. For $K_{\mathrm{in}}$, the row sum must be one. The entries of the matrix $K_{\mathrm{nin}}$ are
$$K_{\mathrm{nin}}^{mn}=\left\{\begin{array}{c} 0\;\mathrm{if}\;K_{\mathrm{in}}^{mn}\neq 0\\ 1\;\mathrm{if}\;K_{\mathrm{in}}^{mn}=0 \end{array}\right..$$
This allows the memory cells with new input to be overwritten and those without to be updated according to how much was read out.

The model described above allows for arbitrary coupling between the memory cells. For this study, we choose $K_{\mathrm{in}}=K_{\mathrm{out}}=\underline{\underline{I}}$, and M=N. This means, at every input cycle, one memory cell is overwritten and one is read out. For this choice of coupling, Equations (Equation 4) and (Equation 5) can be rewritten as
(6)$$x_{out}\left(k+1\right)=G\left(Kx_{out}\left(k-N+1\right)+J\left(k+1\right)\right).$$
We then choose $N=N_{v}+1$ where $N_{v}$ is the number of virtual nodes that will be used for the reservoir computing tasks. This coupling describes a type of ring coupling akin to delay-based reservoir computers with the feedback delay time τ=T+θ, where *T* is the input clock-cycle and θ is the virtual node separation [28,33]. Comparing the continuous and discrete cases gives τ→N, $T\to N_{v}$, and θ→1. We choose such a simple coupling scheme as it has been demonstrated that such coupling topologies perform similarly to random coupling topologies [4]. Using Equation (Equation 6), the rows of the state matrix $\underline{\underline{S}}$ are filled with $N_{v}$ sequential $x_{out}\left(k\right)$, i.e., $S^{k^{\prime },k^{\prime \prime }}=x_{out}\left(N_{v}k^{\prime }+k^{\prime \prime }\right)$ and the bias $S^{k^{\prime },\left(N_{v}+1\right)}=1$ (see Figure 1 for an illustration).

For the nonlinearity, we choose
(7)$$G\left(x\right)=\frac{g_{0}x}{1+x/P_{\mathrm{sat}}},$$
which describes the input response of a semiconductor optical amplifier [34,35].

### 2.3. Input and Mask

The reservoir input is given by a task-dependent time series and a time-delayed version of this time series. Before the data are fed into the reservoir, masks are applied to both input series. The masks consist of $N_{v}$ values drawn from a uniform distribution between 0 and 1. The final input is then given by
(8)$$J\left(k\right)=G_{1}I\left(k^{\prime }\right)M_{1}\left(k^{\prime \prime }\right)+G_{2}I\left(k^{\prime }-d\right)M_{2}\left(k^{\prime \prime }\right)+J_{0},$$
where $G_{1}$ and $G_{2}$ are the input scaling factors, $I\left(k^{\prime }\right)$ is the input time series, *d* is the input delay, $M_{1}\left(k^{\prime \prime }\right)$ and $M_{2}\left(k^{\prime \prime }\right)$ are $k^{\prime \prime }$th entries of the $N_{v}$ dimensional masking vectors, and $J_{0}$ is a constant offset. A sketch of the masked input sequence is shown in Figure 2.

### 2.4. Time Series Prediction Tasks

#### 2.4.1. Mackey–Glass

The Mackey–Glass equation is a delay differential equation which exhibits chaotic dynamics. The reservoir computing benchmarking task is to predict the time series *s* number of steps ahead in the chaotic regime. The Mackey–Glass equation is [36]:(9)$$\frac{dx}{dt}=\beta \frac{x\left(t-\tau \right)}{1+x{\left(t-\tau \right)}^{n}}-\gamma x.$$
We use the standard parameters: τ=17, n=10, β=0.2, and γ=0.1. To create the input sequence $I\left(k^{\prime }\right)$, the time series generated by Equation (Equation 9) is sampled with a time step of dt=1. The corresponding target sequence is then given by $I\left(k^{\prime }+s\right)$.

#### 2.4.2. NARMA10

NARMA10 is a commonly used benchmarking task that is defined by the iterative formula
(10)$$A_{n+1}=0.3A_{n}+0.05A_{n}\left(\sum_{i=0}^{9}A_{n-i}\right)+1.5u_{n-9}u_{n}+0.1,$$
where $u_{n}$ are identically and independently drawn random numbers from a uniform distribution in the interval [0, 0.5] [37]. The reservoir input sequence $I\left(k^{\prime }\right)$ is given by the sequence of $u_{n}$ and the target sequence is given by the corresponding $A_{n}$.

#### 2.4.3. Lorenz

The Lorenz system [38] is given by
(11)$$\begin{array}{c} \frac{dx}{dt}=c_{1}y-c_{1}x,\;\frac{dy}{dt}=x\left(c_{2}-z\right)-y,\;\mathrm{and}\;\frac{dz}{dt}=xy-c_{3}z. \end{array}$$
With $c_{1}=10$, $c_{2}=28$ and $c_{3}=8/3$, this system exhibits chaotic dynamics. We use the *x* variable, sampled with a step size of dt=0.02, as the input $I\left(k^{\prime }\right)$ for two time series prediction tasks. The first is one step ahead (s=1) prediction of the *x* variable. The second is one step ahead (s=1) cross-prediction of the *z* variable.

### 2.5. Simulation Conditions

For all tasks, the reservoir is initialised with an input sequence $I\left(k^{\prime }\right)$ of length 10,000. The system is then trained on $K_{\mathrm{tr}}=$ 10,000 inputs. This is followed by another buffer of 10,000 inputs, before the performance is tested on a sequence of $K_{\mathrm{te}}=5000$ inputs, unless stated otherwise. For each task, the reservoir parameters are kept identical and are as given in Table 1. The input scaling of the primary input $G_{1}$ (nondelayed input) and the offset $J_{0}$ are scaled such that the input range for each task is approximately [0.4, 1.3]. The scaling of the delayed input $G_{2}$ and the input delay *d* are used as the optimisation parameters. For each task, the performance is averaged over 100 realisations of the random masks and also, in the case of NARMA10, the random inputs.

## 3. Results

The performance of the reservoir with additional delayed input is tested on six tasks; we first consider Mackey–Glass time series prediction for one, three, and ten steps into the future. The results of the Mackey–Glass tasks, and their relation to the delayed input parameters, are depicted in Figure 3a–c. By inspecting the evolution of the performance error as a function of *d* and $G_{2}$, an optimal performance and thus an optimal value for *d* can be identified (brightest light yellow region). This value, however, depends on the task and thus changes in between the panels. In order to quantify the impact of the delayed-input strength on the performance, we present scans of the delayed-input strength $G_{2}$ for the optimal input delay *d*, for each task, in Figure 4a–c. $G_{2}=0$ corresponds to the system without delayed input and should be used as the reference to quantify the performance boost due to the delayed input. For each of the three cases, the delayed input leads to a reduction in the NRMSE, ranging from 20% for s=1 to over a factor three for s=10. The optimal values for the delay and the input scaling $G_{2}$ vary depending on the number of steps *s* predicted into the future. In agreement with the results presented in [39], larger input scaling is required as *s* increases, indicating that nonlinear transforms become increasingly important. In terms of the absolute performance, similar results are achieved compared with other studies [39,40], despite the number of virtual nodes used in this study being significantly lower.

The results for the NARMA10 task are shown in Figure 3d and Figure 4d. Without the delayed input ($G_{2}=0$) the performance of the reservoir is very poor. This is in contrast to the Mackey–Glass s=1 for which the performance without delayed input (Figure 3a with $G_{2}=0$) is reasonable. Moreover, this finding supports the general observation that reservoir computers have to be optimised to individual tasks and perform poorly as universal approximators [1,10,11]. The inclusion of delayed input significantly reduced the NARMA10 error, reaching an NRMSE of about 0.3 for the input delay d=9. In absolute terms, an NRMSE of 0.3 is within the range of typically quoted best values (NRMSE = 0.15–0.4) [4,10,41,42,43,44], however, it is usually achieved with a much higher output dimension than the $N_{v}=30$ used here. The performance achieved in this study came at a very low computational cost. As a comparison, in [44] the authors investigate the influence of combining echo state networks with different timescales and achieve a best performance of just under 0.4 for the NRMSE, at a greater computational expense.

The remaining two tasks are one step ahead Lorenz *x* prediction and one step head Lorenz *z* cross-prediction, the results of which are shown in Figure 3e,f and Figure 4e,f. In both cases there is an improvement in the performance with the correct choice of the delayed input. It is has been demonstrated that the Lorenz *x* one step ahead prediction requires only the very recent history of the *x*-variable time series [13], and we find the optimal input delay of d=1 to be consistent with this prior knowledge. For the Lorenz *z* cross-prediction task, on the other hand, there is a strong dependence on the history of the Lorenz *x* variable. In this case, the best performance is achieved when the second input is delayed by d=14 time steps. The optimal delayed-input scaling $G_{2}$ is larger for the Lorenz *z* task than the Lorenz *x* task (as seen by comparing the positions of the minima in Figure 4e,f), indicating that the cross-prediction task requires a greater degree of nonlinearity as well as a longer memory.

In order to demonstrate that the improvement in the performance with delayed input is not specific to the reservoir parameters used for Figure 3, in Figure 5 we show the NRMSE for the Mackey–Glass s=10 task as a function of (a) the virtual node coupling strength *K* and (b) the coupling delay *N* (i.e., the number of memory cells). These parameters have a strong influence on the properties of the reservoir. In both cases the NRMSE without delayed input (orange dotted line) shows a large variation over the respective parameter ranges and is always larger than the error with optimised delayed input (blue dashed line). With optimised delayed input the variation in the error is comparatively small, demonstrating that the inclusion of the delayed input works well independent of the reservoir properties. The peak in the NRMSE at N=30 in Figure 5b is a well-known resonance effect that occurs at resonances between the number of virtual nodes and the coupling range *N*, equivalent to clock time and delay resonances in time continuous systems [45].

To further demonstrate the universality of this method, we show the NARMA10 error with delayed input for a time continuous reservoir in Figure 6. In this case the reservoir is given by the Stuart–Landau equation with time-delayed feedback (see Appendix B). The reservoir parameters have not been optimised for the NARMA10 task, resulting in very poor performance without delayed input ($G_{2}=0$). With optimised delayed-input parameters reasonable performance is achieved, similar to the optimal results for the memory cell reservoir in Figure 3d. For the Stuart–Landau reservoir, optimal performance is achieved for the input delay d=10, whereas, for the memory cell reservoir, the optimal input delay is d=9. This is because the required input delay depends both on the dynamics of the reservoir as well as the memory requirements of the particular task.

## 4. Discussion

We have shown that, for various time series prediction tasks, including a delayed version of the input can lead to a substantial improvement in the performance of a reservoir. We have demonstrated this using a simple map describing a semiconductor optical amplifier nonlinearity and a ring-like coupling realised via memory cells. With this approach we were able to use one unaltered reservoir to perform well on six different tasks, each with different memory and nonlinear transform requirements. The performance boost due to the delayed input is achieved over a wide range of the reservoir parameters and was also demonstrated for a time continuous system, indicating that our approach is applicable to a wide range of reservoirs.

Our results are significant for a number of reasons. Firstly, we have demonstrated that computationally expensive hyperparameter optimisation can be circumvented by tuning only two input parameters. By including an additional delayed input, reasonable performance can be achieved using an unoptimised reservoir. Nevertheless, we note that, depending on the requirements for a given task, additional hyperparameter optimisation may be necessary. Secondly, to the best of our knowledge, this is the first demonstration of an identical reservoir performing well on such a large range of tasks. Thirdly, the simplicity of our approach means that it is well suited to be applied on physical reservoirs.

This study has raised several questions surrounding delay-based reservoir optimisation that require further investigation. For example, the optimal delayed-input parameters are task dependent and how these relate to a given task is not fully understood. The NARMA10 results presented in this study indicate that the optimal delayed-input parameters are related both to the reservoir and requirements of the task. This means that it may be possible to not only use reservoir computing for real-world time series prediction tasks, but also to gain insights into the dynamical systems being investigated. For example, in tasks such as El Niño prediction where the underlying dynamical system is very complex and the relevant physical processes are not fully understood [46]. Here, investigations surrounding delay-based input could provide critical insight into the involved timescales. Furthermore, the minimum requirements for a reservoir to yield good performance on a range of tasks by only tuning the delayed input parameters remain to be determined.

A natural extension of our proposed approach is to include multiple delayed input terms. This would bring the reservoir computing approach closer to classical statistical forecasting methods such as NVAR and could lead to a further improved performance, especially for tasks involving multiple disparate timescales. However, possible performance improvement with added input terms must be weighed against the associated increase in the computational cost as each added input adds two new optimisation parameters.

## Figures and Tables

**Figure 1 entropy-23-01560-f001:**
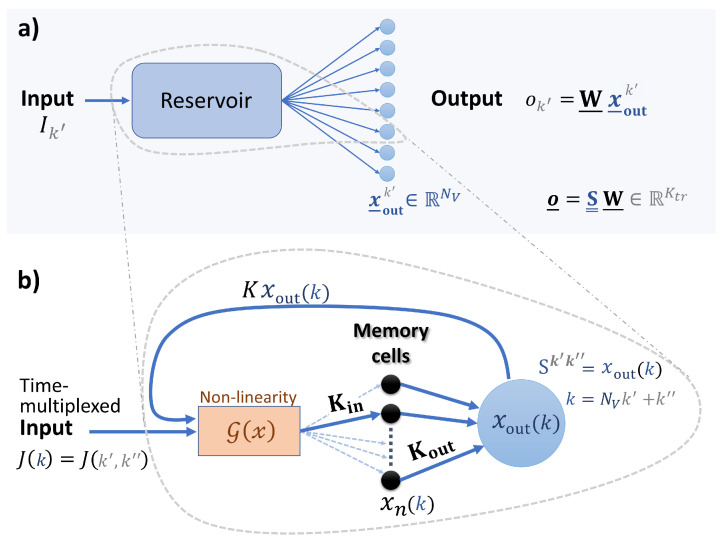
(**a**) Sketch of the reservoir computing concept. The vector ${\underline{x}}_{\mathrm{out}}^{k^{\prime }}$ is the responses of the reservoir to an input $I_{k^{\prime }}$ and the corresponding output $o_{k^{\prime }}$ is generated by a weighted sum of these responses. The read-out weights $\underline{W}$ are trained. (**b**) Sketch of the memory cell reservoir described in Section 2.2, where a time-multiplexed input J(k) is fed into the reservoir ($k=N_{v}k^{\prime }+k^{\prime \prime }$, please see Figure 2 for the construction of the time-multiplexed input). The index *n* labels the memory cells (in total *N*) that are addressed via the coupling matrices $K_{\mathrm{in}}$ and $K_{\mathrm{out}}$. *K* labels the feedback strength at which the output of the memory cells $x_{\mathrm{out}}\left(k\right)$ (given by Equation (Equation 5)) is fed back into the nonlinearity G. The elements of the state matrix $\underline{\underline{S}}$ are given by $S^{k^{\prime },k^{\prime \prime }}=x_{\mathrm{out}}\left(N_{v}k^{\prime }+k^{\prime \prime }\right)$, with $k^{\prime }\in \left(1\cdots K_{tr}\right)$ and $k^{\prime \prime }\in \left(1\cdots N_{v}\right)$, i.e., one row of $\underline{\underline{S}}$ corresponds to the vector of responses ${\underline{x}}_{\mathrm{out}}^{k^{\prime }}$ in (**a**).

**Figure 2 entropy-23-01560-f002:**
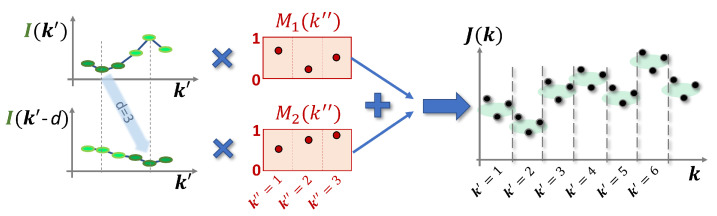
Sketch of the generation of the final time-multiplexed input sequence $J\left(k\right)$ using the task-dependent input $I\left(k^{\prime }\right)$, a delayed version of this input $I\left(k^{\prime }-d\right)$, and the masks $M_{1}\left(k^{\prime \prime }\right)$ and $M_{2}\left(k^{\prime \prime }\right)$, as described by Equation (Equation 8).

**Figure 3 entropy-23-01560-f003:**
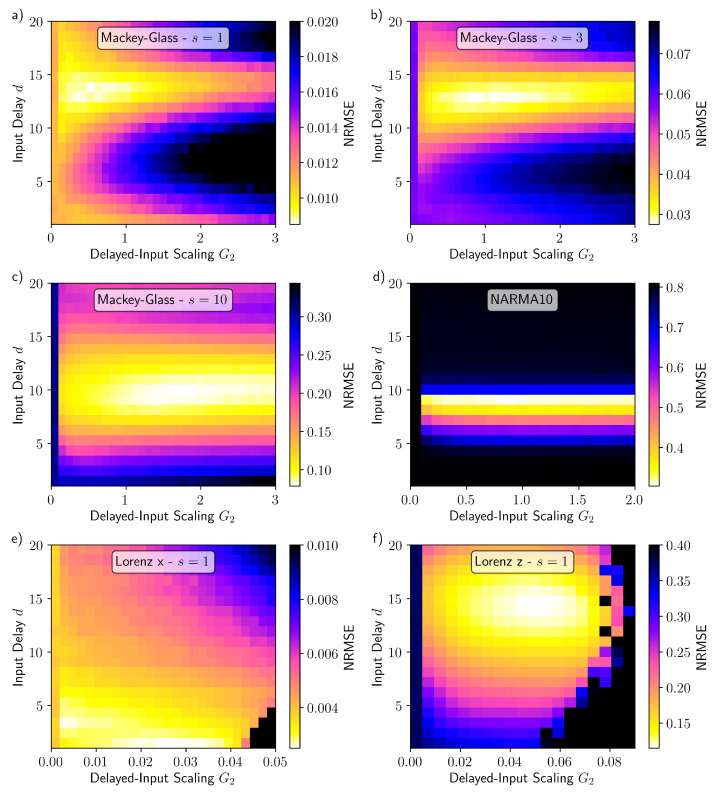
NRMSE as a function of the delayed input parameters *d* and $G_{2}$ for Mackey–Glass (**a**) one, (**b**) three, and (**c**) ten steps ahead prediction, (**d**) NARMA10, (**c**) Lorenz *x* one step ahead prediction, and (**f**) Lorenz *z* one step ahead cross-prediction. Parameters are as stated in Section 2.5, except for (**a**,**e**) where $K_{\mathrm{te}}=30,000$.

**Figure 4 entropy-23-01560-f004:**
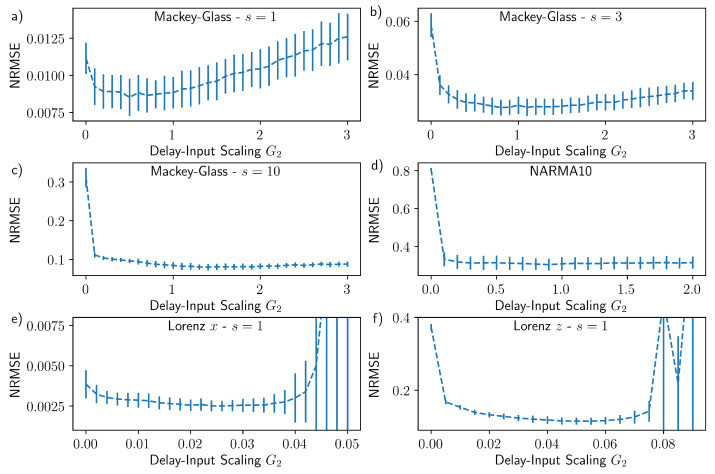
NRMSE for optimised input delay *d*, as a function of the delayed-input scaling $G_{2}$ for Mackey–Glass (**a**) one, (**b**) three, and (**c**) ten steps ahead prediction, (**d**) NARMA10, (**c**) Lorenz *x* one step ahead prediction, and (**f**) Lorenz *z* one step ahead cross-prediction. The error bars indicate the standard deviation. The optimal input delays are (**a**) d=14, (**b**) d=13, (**c**) d=9, (**d**) d=9, (**e**) d=1, and (**f**) d=15. The remaining parameters are as stated in Section 2.5, except for (**a**,**e**) where $K_{\mathrm{te}}=30,000$.

**Figure 5 entropy-23-01560-f005:**
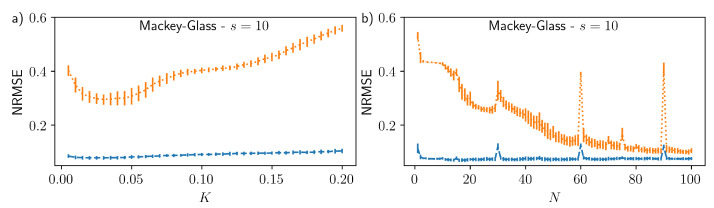
NRMSE for Mackey–Glass 10 step ahead prediction as a function of (**a**) the virtual node coupling strength *K* and (**b**) the coupling delay *N*. The orange dotted (blue dashed) lines show the results without (with) delayed input. Along the blue curve the delayed input parameters *d* and $G_{2}$ have been optimised (see Figure A1 in Appendix A for their values). The error bars indicate the standard deviation. All remaining parameters are as stated in Section 2.5.

**Figure 6 entropy-23-01560-f006:**
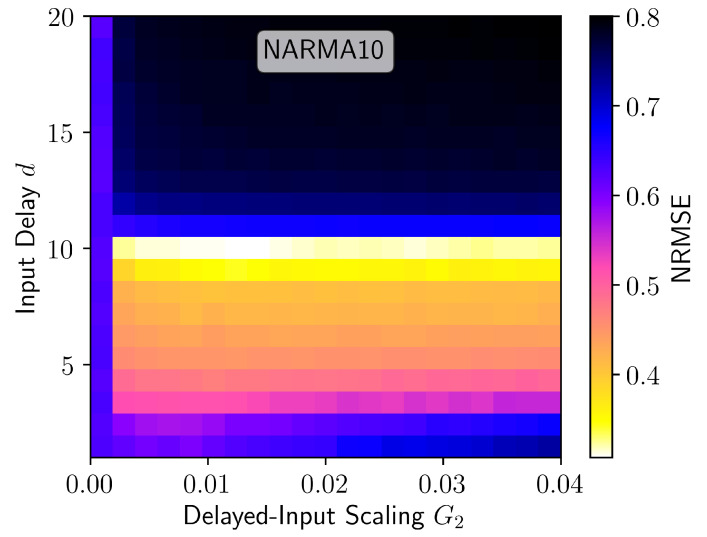
NRMSE for the NARMA10 task as a function of the delayed input parameters *d* and $G_{2}$ using the Stuart–Landau delay-based reservoir computer described in Appendix B.

**Table 1 entropy-23-01560-t001:** Reservoir and input parameter values.

Parameter	Value	Parameter	Value
$g_{0}$	40	$P_{\mathrm{sat}}$	1
*K*	0.02	*N*	31
$N_{v}$	30	λ	5 × 10${}^{-6}$
$G_{1}$ (Mackey–Glass)	1	$J_{0}$ (Mackey–Glass)	0
$G_{1}$ (Lorenz)	0.03	$J_{0}$ (Lorenz)	0.85
$G_{1}$ (NARMA10)	1.8	$J_{0}$ (NARMA10)	0.4

## Data Availability

The datasets generated and analysed during the current study are available from the corresponding author on reasonable request.

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
