# Peer review of "Reservoir Computing with Delayed Input for Fast and Easy Optimisation"

_entropy, 2021, doi:10.3390/e23121560_

Round 1
Reviewer 1 Report
The authors demonstrate an approach that reduces the need for hyper parameter optimization and is well suited to boosting the performance of physically implemented reservoir computers.
- The author needs to explain the parameters in Figure 1. In Figure 1 (b), what type of input does J(k) represent? What does k stand for? What is the relationship between k and Kin and Kout? Please explain in detail.
- In equation (3), what does mean?
- In equation (4) and equation (5), what does the Kin, Knin and Kout parameters contain the subscripts m and n mean? Is it related to the dimension M x N of the coupling matrix? Please supplement.
- In Section 3, the reservoir performance of delayed input is tested on six tasks. Firstly, the time series prediction of Mackey-Glass-s= 1, 3 and 10 steps is considered. Why do you choose the step size of 1, 3 and 10? Is the step size of 1, 3 and 10 representative and reasonable for the experimental study?
- Section 4 of the paper states that the optimal delay input parameters depend on the task, and the relationship between these parameters and the given task has not been fully understood. What is the criterion for determining the optimal delay input parameters?
Author Response
Point by point Reply
Reviewer 1: The authors demonstrate an approach that reduces the need for hyper parameter optimization and is well suited to boosting the performance of physically implemented reservoir computers.
- The author needs to explain the parameters in Figure 1. In Figure 1 (b), what type of input does J(k) represent? What does k stand for? What is the relationship between k and Kin and Kout? Please explain in detail.
Answer: We included the explanation of the parameters in the caption.
- In equation (3), what does mean?
Answer: The performance error is measured by the NRMSE. It is normalized to the variance of the input sequence (var(o)). This way the error gives 1 if the prediction error scatters as much as the input itself.
- In equation (4) and equation (5), what does the Kin, Knin and Kout parameters contain the subscripts m and n mean? Is it related to the dimension M x N of the coupling matrix? Please supplement.
Answer: We are sorry for the confusion. Kin, Knin and Kout are matrices and the subscripts describe the components, thus they should not appear bold in the equation.
- In Section 3, the reservoir performance of delayed input is tested on six tasks. Firstly, the time series prediction of Mackey-Glass-s= 1, 3 and 10 steps is considered. Why do you choose the step size of 1, 3 and 10? Is the step size of 1, 3 and 10 representative and reasonable for the experimental study?
Answer: We thank the referee for this question. Actually we chose them to have one representative of a long, a shot and an intermediate prediction time s. For an experimental realization all values are reasonable. The specific choice will depend on the question that the reservoir computer is supposed to solve (e.g. if real weather data are supposed to be predicted s=10 or s=1 lead to long term or a short term predictions).
- Section 4 of the paper states that the optimal delay input parameters depend on the task, and the relationship between these parameters and the given task has not been fully understood. What is the criterion for determining the optimal delay input parameters?
Answer: The performance error was the quantity that we used to determine the optimal delay (brightest yellow areas in Fig3). We added a paragraph to make that more clear.
Reviewer 2 Report
This is a very interesting manuscript, which deals with a very important and timely problem: use of ML to predict time series of dynamical systems.
Reservoir Computing has indeed been identified as one of the most promising approaches to deal with this ill-defined problem.
Overall the manuscript is well written, the literature review comprehensive, and the research extensive.
I am very supportive of this work. There are a couple of minor typos, which can easily be fixed.
However, there is one methodological point, which I hope the authors could discuss. Attention Networks (AN) have recently been identified as an alternative to Reservoir Computing. I would very much appreciate if the authors could comment on why they chose RC over AN, and if would be the advantage of RC over AN, if any.
Author Response
Reviewer2: This is a very interesting manuscript, which deals with a very important and timely problem: use of ML to predict time series of dynamical systems.
Reservoir Computing has indeed been identified as one of the most promising approaches to deal with this ill-defined problem. Overall the manuscript is well written, the literature review comprehensive, and the research extensive.
I am very supportive of this work. There are a couple of minor typos, which can easily be fixed.
Answer: Thank you for the positive evaluation of our work. We checked for typos and fixed the mistakes.
However, there is one methodological point, which I hope the authors could discuss. Attention Networks (AN) have recently been identified as an alternative to Reservoir Computing. I would very much appreciate if the authors could comment on why they chose RC over AN, and if would be the advantage of RC over AN, if any.
Answer: In the context of RC we are not aware of how the attention mechanism, which is usually used in transformer networks, can be beneficial. Since we are not experts in that respect we did not discuss this matter in the manuscript. Maybe the referee could provide us with some literature for further reading related to this topic.